# Surface single-molecule dynamics controlled by entropy at low temperatures

J.C. Gehrig[1,*], M. Penedo[1,*], M. Parschau[1], J. Schwenk[1], M.A. Marioni[1], E.W. Hudson[1,2] & H.J. Hug[1,3]

Configuration transitions of individual molecules and atoms on surfaces are traditionally described using an Arrhenius equation with energy barrier and pre-exponential factor (attempt rate) parameters. Characteristic parameters can vary even for identical systems, and pre-exponential factors sometimes differ by orders of magnitude. Using low-temperature scanning tunnelling microscopy (STM) to measure an individual dibutyl sulfide molecule on Au(111), we show that the differences arise when the relative position of tip apex and molecule changes by a fraction of the molecule size. Altering the tip position on that scale modifies the transition's barrier and attempt rate in a highly correlated fashion, which results in a single-molecular enthalpy-entropy compensation. Conversely, appropriately positioning the STM tip allows selecting the operating point on the compensation line and modifying the transition rates. The results highlight the need to consider entropy in transition rates of single molecules, even at low temperatures.

[1] Empa, Swiss Federal Laboratories for Materials Science and Technology, CH-8600 Dübendorf, Switzerland. [2] Department of Physics, Pennsylvania State University, University Park, Pennsylvania 16802, USA. [3] Department of Physics, University of Basel, CH-4056 Basel, Switzerland. * These authors contributed equally to this work. Correspondence and requests for materials should be addressed to M.A.M. (email: miguel.marioni@empa.ch).

Observing transitions of molecules between metastable states provides insight into the dynamics underpinning many catalytic[1,2] and biological processes[3–8]. We can restrict the phase space of the molecules by adsorbing them on a surface such that their transition kinetics becomes accessible to STM. Thus for instance, it is possible to calculate hopping rates from series of images tracking the surface motion of Al atoms on Al(111)[9], tetrapyridylporphyrin on Cu(111)[10], tetraphenylporphyrin monomers and dimers on Cu(111)[11] and cobalt(II) octaethylporphyrin on graphite[12]. Certain adsorbed molecules can also pivot around their bond to the substrate, and distinct rotational states can be imaged in sequence by STM, as shown for $O_2$ on Pt(111)[13], Cu phthalocyanine on $C_{60}$ (refs 14,15) and zinc tetra-(3,5-di-tert-butylphenyl)porphyrin on Ag(100)[16]. In these types of transitions the rate of occurrence can also be obtained by counting individual transitions from telegraph noise in the current signal[13,17,18]. Such rate data is the basis for studying the temperature dependent physics of the transition, most commonly described in terms of an Arrhenius equation[19] when the process is thermally activated. In this context one can determine energy barriers to the transitions ($E^*$) and transition attempt rates ($A^*$)[19].

The latter parameter is formally a constant, yet in various instances the measured values of $A^*$ depart from expected behaviour in two ways: first, $A^*$ turns out to be consistently different from molecular vibration frequencies in realistic adsorption surface potentials[20–22]; Second, the values for the same system and experiment conditions can vary by orders of magnitude, notably in dibutyl sulfide (DBS) on Au(111)[21,23]. In this work we look closer at the reasons for this puzzle, using an STM tip to modify and characterize transitions rates of a single-molecule model system.

Over the area where the transitions are detected by the STM, we find from comparing tip position-dependent transition rate data at different temperatures that between equilibrium and transition states the enthalpy- and entropy-differences are not uniform. Their values display a correlated structure as function of tip position that does not reproduce the symmetry of the substrate entirely. Nevertheless, enthalpy and entropy are compensated for the molecule, implying an isokinetic temperature of $9.1 \pm 0.2$ K. In particular the variation of the transition entropy difference with the position of the tip can result in order of magnitude variations of the pre-exponential factor in Arrhenius-law descriptions of the process. Our work shows that entropic contributions to the dynamics of atoms and molecules during surface transitions and manipulation cannot be disregarded, even at low temperatures, and gives an example of how STM could be used to measure and modify the entropy of atomic or molecular-scale systems and their interactions with tip and substrate. The STM tip can be used at a level of interaction with the molecule where the molecular transitions are still thermally activated, but the barriers are modified by the tip. Thus different regimes of the thermally activated dynamics of the transitions can be probed by appropriately selecting the tip position, but the compensation itself is independent of the means by which the barrier height is modified. It is therefore important that future models for describing the transition dynamics of surface-adsorbed molecules at low temperatures account for the details of the excitation process and their effect on entropy.

## Results

**Molecule images**. We study the Arrhenius behaviour of a dibutyl sulfide molecule (DBS) adsorbed on Au(111) in the temperature range 5–10 K. These molecules adsorb via their central sulfur atom, which becomes the pivot for molecular rotation, with the two butyl groups hopping between equivalent sites on the Au(111) surface.

STM measurements at 5.44 K reveal isolated molecules appearing as linear objects of about 120 pm height (Fig. 1a). Their rotation by hopping can be induced with temperature (Fig. 1d–g) or inelastic tunnelling[21]. Note that the latter mechanism would trigger molecule responses at a temperature-independent rate. In this work we use a bias of 200 mV, which is well below the 375 mV threshold for the inelastic tunnelling that was shown to excite a C–H stretching mode[23] in DBS on Au(111), leading to rotational hopping of the molecule.

**Transition rates**. Consistent with a bias significantly below the threshold for inelastic tunnelling the count of rotation hops of DBS have a strong temperature dependence. They rarely take place at temperatures below 6 K, with a hopping rate sufficiently small for the butyl groups to remain in the same position during the recording time of an STM image, as shown in Fig. 1c,d. As the hops become more frequent with temperature increasing from 5.41 to 13.60 K (Fig. 1d–g), the butyl groups occupy more than

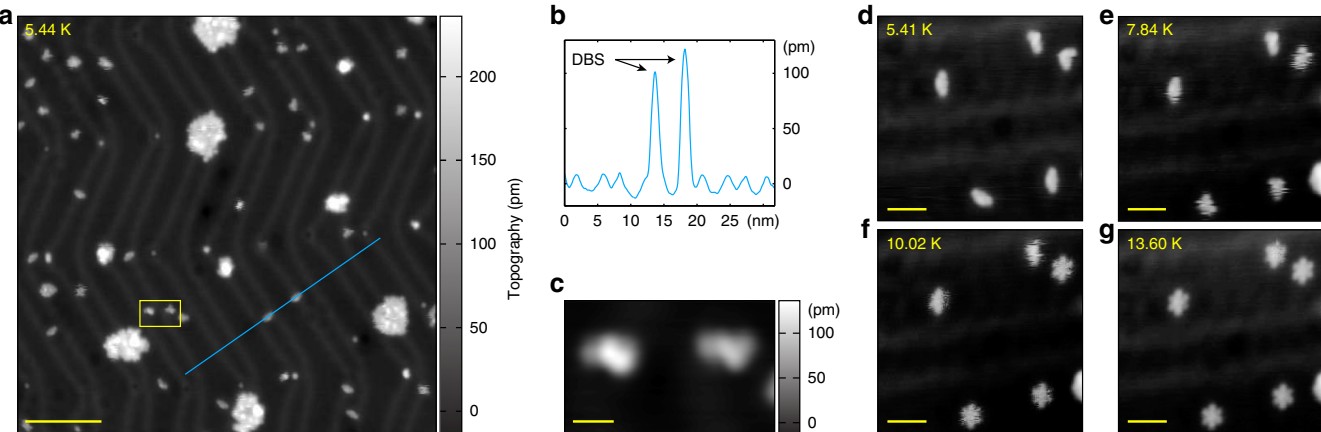

**Figure 1 | STM measurements of dibutyl sulfide molecules. (a)** Overview image of dibutyl sulfide molecules on Au(111). Isolated molecules decorate the surface. Scale bar, 10 nm. (**b**) Section of (**a**) along the blue line, showing the height of the DBS molecule (distinct, large peaks) as well as the typical Au(111) herringbone structure (periodic, small peaks). (**c**) Scan of the $5.2 \times 3.3$ nm$^2$ yellow-framed area of (**a**). Scale bar, 1 nm. (**d–g**) Series of topographs of isolated molecules taken at increasing temperatures. Scale bar, 2 nm. At 5.41 K molecules appear to be linear, whereas at 13.60 K six lobes are visible consistent with the occupation of the different rotated configurations.

one position during the image scan, distorting the linear appearance of the molecules. At 13.60 K the butyl groups hop so quickly that the molecule appears to have a hexagonal shape (Fig. 1g).

In Fig. 2b–g we show representative hopping-rate measurements made on the left molecule of Fig. 1c. The figure highlights both a general trend of hopping rate increasing with temperature and the rate's strong (one order of magnitude) dependence on the position of the STM tip relative to the molecule. The rate patterns are not random, and lack the symmetry of the highest-temperature images (for example, 13.6 K image in Fig. 1g, acquired on a different dibutyl sulfide molecule with a different tip).

To appreciate the meaning of these observations a previous result from Baber et al.[21] is useful. It was found (cf. Fig. 4 of their work) that at specific positions of the tip close to the outer rim of the hopping DBS molecule, the transitions between the three nearly equivalent surface states could be identified. In our work,

the tip does not remain stationary at such positions but is scanned, so that tunnelling current error jumps are recorded at each tip position within the surface area probed by the rotating molecule (provided the temperature is high enough). For each tip position it is not practical to disaggregate hops of the molecule between different pairs of equilibrium states, and we therefore only register the total count. (The energy barrier extracted from the hopping rate is thus an effective energy barrier.) But so long as the tip does not affect the transitions, the recorded rate should be the same for all tip positions covering the transition between pairs of equilibrium states.

**Influence of the STM tip.** Figure 2, and particularly the data taken at temperatures above 8.38 K (Fig. 2e–g) measure non-zero transition counts over an area of roughly hexagonal shape, but the counts depart strongly from a uniform value. This indicates that contrary to the initial assumption the tip-molecule interaction

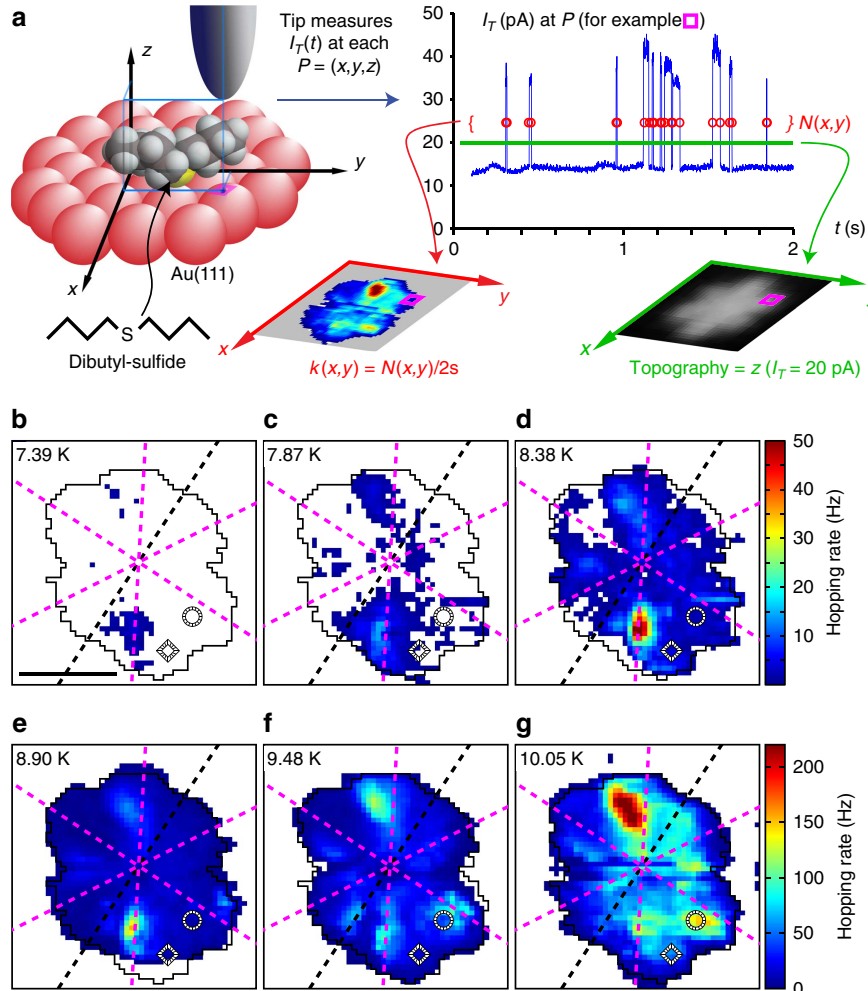

**Figure 2 | Hopping rate measurement and analysis.** (**a**) Measurement principle: a dibutyl sulfide molecule adsorbed on the Au(111) surface (top left) is scanned at a rate of 400 s per line (both directions). $z$-feedback keeps the average tunnelling current constant at 20 pA and results in the topography image (bottom right). Telegraph noise recorded at a high measurement bandwidth (top right) allows identifying and counting the $N$ molecular hops occurring over the 2 s span that the tip spends over each position $(x, y)$, and results in the hopping rate $k(x, y, T)$ (bottom left). (**b–g**) Maps of the hopping rate $k(x, y, T)$ highlight the influence of both temperature and relative molecule-tip position $(x, y)$. Scale bar, 1 nm. Note that in these and all subsequent spatial maps, the colour white indicates regions where no hopping was discerned using this technique. The black/white striped ring and diamond markers identify two specific positions $r_1$ and $r_2$ for further analysis in Fig. 3. The dashed black line is parallel to the nearby herring bone ridge, while the dashed magenta lines indicate the symmetry axes (rows of atoms) of the Au(111) surface. The black contour outlines the region utilized for further analysis, containing non-zero rates for at least three temperatures. The hexagonal shape of this contour is reminiscent of the typical hexagonal shaped appearance of STM topography map of the molecule typically occurring at higher temperatures (Fig. 1g).

modifies the energy landscape of the surface. Further experiments (not shown) revealed that the rate patterns also depend on the tip's average height over the molecule. Therefore, we can trace the origin of the lack of symmetry of the patterns to the structure of the tip with which the molecule interacts.

The consequence of these considerations is that, *a priori*, we must treat the tip as an integral part of the system, inasmuch as it can modify the thermodynamic potentials. However, these tip-induced modifications could be used to extract information of the system dynamics, as we will show.

To assess the extent of the tip-influence and establish the implications for the dynamics of molecules on surfaces in general, we analyse how the hopping rate $k(x, y, T)$ changes with temperature $T$ at each $(x, y)$ tip-position. After ensuring the rate maps measured at each temperature are in registry we can carry out the comparison as indicated schematically in Fig. 3a. Examples from selected positions in Fig. 3b indicate that the behaviour is phenomenologically described by an Arrhenius equation,

$$k(x, y, T) = A^*(x, y) \cdot \exp\left(-\frac{E^*(x, y)}{k_B T}\right), \quad (1)$$

where $k_B$ is Boltzmann's constant, $E^*(x, y)$ is an apparent energy barrier height, and $A^*(x, y)$ is an apparent attempt rate. $E^*(x, y)$ and $A^*(x, y)$ describe the lines fitting the data in Fig. 3b and are given inside that figure's panel.

For the complete set of scanned $(x, y)$ positions the $E^*(x, y)$ and $A^*(x, y)$ fitting the measured $k(x, y, T)$ with equation (1) results in maps depicted in Fig. 3c,d, respectively. Panels e and f indicate the corresponding errors. Note that the patterns of the energy barrier, $\log_{10}(A^*/\text{Hz})$, and those of the corresponding errors are plotted with the same colour scales, respectively. The fact that the energy barrier- and $\log_{10}(A^*/\text{Hz})$-lobe patterns appear with colours ranging from red (high value) to blue (low value), while the corresponding error maps show weak variation in the dark blue colour range shows that the determined energy barriers and $\log_{10}(A/\text{Hz})$-values, and the observed variations of their patterns are statistically significant. (Panel g indicates the number of points used for the fit.) For most tip positions the Arrhenius equation is a good fit to the data, meaning that the transitions are essentially thermally activated.

## Discussion

Several observations are noteworthy. First, the effective energy barrier height $E^*(x, y)$ spans an unexpectedly large range from a few meV to about 20 meV (Fig. 3c). Second, as we found with the rate, also the energy barrier depends on tip-position, showing that the overall energy landscape probed by the adsorbed molecule is strongly influenced by the presence of the tip. And third, the attempt rate $A^*(x, y)$ varies over 10 orders of magnitude over the scanned relative positions, but its map (Fig. 3d) exhibits a close resemblance to that of $E^*(x, y)$ (Fig. 3c), suggesting a fundamental underlying connection between the two.

Taking all pairs of $\ln(A^*(x, y))$ and $E^*(x, y)$ and plotting them in Fig. 3h reveals a clustering of the data blue diamonds on a straight line. Hexagons indicate the location on the plot of data from literature for rotation of dibutyl sulfide on Au(111), and the striped markers correspond to the lines presented in Fig. 3b.

Taking $A^*$ in Hz we can fit the data in Fig. 3h with the line

$$\ln A^*(x, y) = \frac{E^*(x, y)}{E_{MN}} + \ln A_{00}, \quad (2)$$

where the parameters are the energy $E_{MN} = 0.78 \pm 0.02$ meV and $\ln A_{00} = 2.5 \pm 0.3$.

Equation (2) expresses for the single dibutyl sulfide molecule what is known for example, for families of related chemical reactions[24] as the Cremer-Constable- or Meyer-Neldel rule (see refs 1,2 for a review). We can see that the role of the STM tip is to modify the conditions in which the transitions take place. Specifically, for each relative tip-molecule position the DBS rotation transition encounters a different energy barrier height (which is connected to the attempt rate). But the transition itself is not barrier-less and is thermally activated. Replacing $A^*(x, y)$ in equation (1) by the expression in equation (2) reveals that at $T = T_{iso} = E_{MN}/k_B = 9.1 \pm 0.2$ K the rates become position-independent, that is, $k(x, y) = \ln A_{00}$. It has been pointed out by Boisvert et al.[24] that the Meyer–Neldel energy is expected to be of the order of the energy excitations available in the energy reservoir.

The variability of $A^*(x, y)$ over orders of magnitude becomes explicitly apparent by writing equation (1) in terms the canonical multidimensional transition state theory result[25],

$$k(x, y, T) = \frac{k_B T}{h} \exp\left(\frac{\Delta S^{\ddagger}(x, y)}{k_B}\right) \exp\left(-\frac{\Delta H^{\ddagger}(x, y)}{k_B T}\right) \quad (3)$$

where $\Delta H^{\ddagger}(x, y)$ is the tip-position dependent enthalpy difference between the transition- (metastable) and ground state enthalpy. Similarly $\Delta S^{\ddagger}(x, y)$ is the entropy difference, which can be written in terms of the partition function $Z^{\ddagger}(x, y)$ ($Z_0(x, y)$):

$$\Delta S^{\ddagger}(x, y) = k_B \ln\left(\frac{Z^{\ddagger}(x, y)}{Z_0(x, y)}\right). \quad (4)$$

In this context the exponential prefactor then takes the form

$$A^*(x, y, T) = \frac{k_B T}{h} \exp\left(\frac{\Delta S^{\ddagger}(x, y)}{k_B}\right). \quad (5)$$

which accounts for the weak temperature dependence of $A^*$. Equation (3) can be shown to be equivalent to a theory of the escape rate of a particle trapped in a one-dimensional potential and coupled bilinearly to a bath of an infinite set of harmonic oscillators[25]. More specifically we can define an entropy difference between the metastable- and transition state as $\Delta S^{\ddagger}(x, y) = k_B \ln(Z^{\ddagger}(x, y)/Z_0(x, y))$, of which the apparent attempt rate $A^*(x, y)$ depends exponentially.

Thus, using the logarithmic form of equation (3) we can obtain the enthalpy map (Fig. 4a) and entropy map (Fig. 4c). Note that $\Delta S^{\ddagger}(x, y)$ varies between $-2.2$ and $0$ meV K$^{-1}$ and has the same pattern as $\Delta H^{\ddagger}(x, y)$, which expresses an enthalpy–entropy compensation. Yet a better point of comparison is the thermodynamic potential obtained by multiplying $\Delta S^{\ddagger}(x, y)$ with a relevant temperature. For this case we chose the isokinetic temperature $T_{iso} := E_{MN}/k_B = 9.1 \pm 0.2$ K. This amounts to a rescaling of the map in Fig. 4c, as indicated by the right scale bar.

It becomes clear that apart from the similarity of the patterns in Fig. 4a,c, the data $T_{iso}\Delta S^{\ddagger}$ is in the same range as $\Delta H^{\ddagger}$ (see right scale bar of Fig. 4c). Enthalpy and entropy have comparable effects on the dynamics of the DBS molecule rotation on the Au(111) surface.

Marbach et al.[26] suggested that entropy can be important in such a process, but also pointed out that its contribution should become negligible at low temperatures. The latter statement does not contradict our finding, but calls for a more careful assessment of the sources of entropy. Experimental evidence suggests that configurational entropy, increasing with the number of degrees of freedom, could be behind the close proximity of the temperatures at which dialkyl-sufides with different chain lengths rotate[21]. Accordingly, if the STM tip constrained the possible locations of the butyl chains of the DBS molecule (analogously to the lateral confinement of metal-organic meshes in the work of Palma

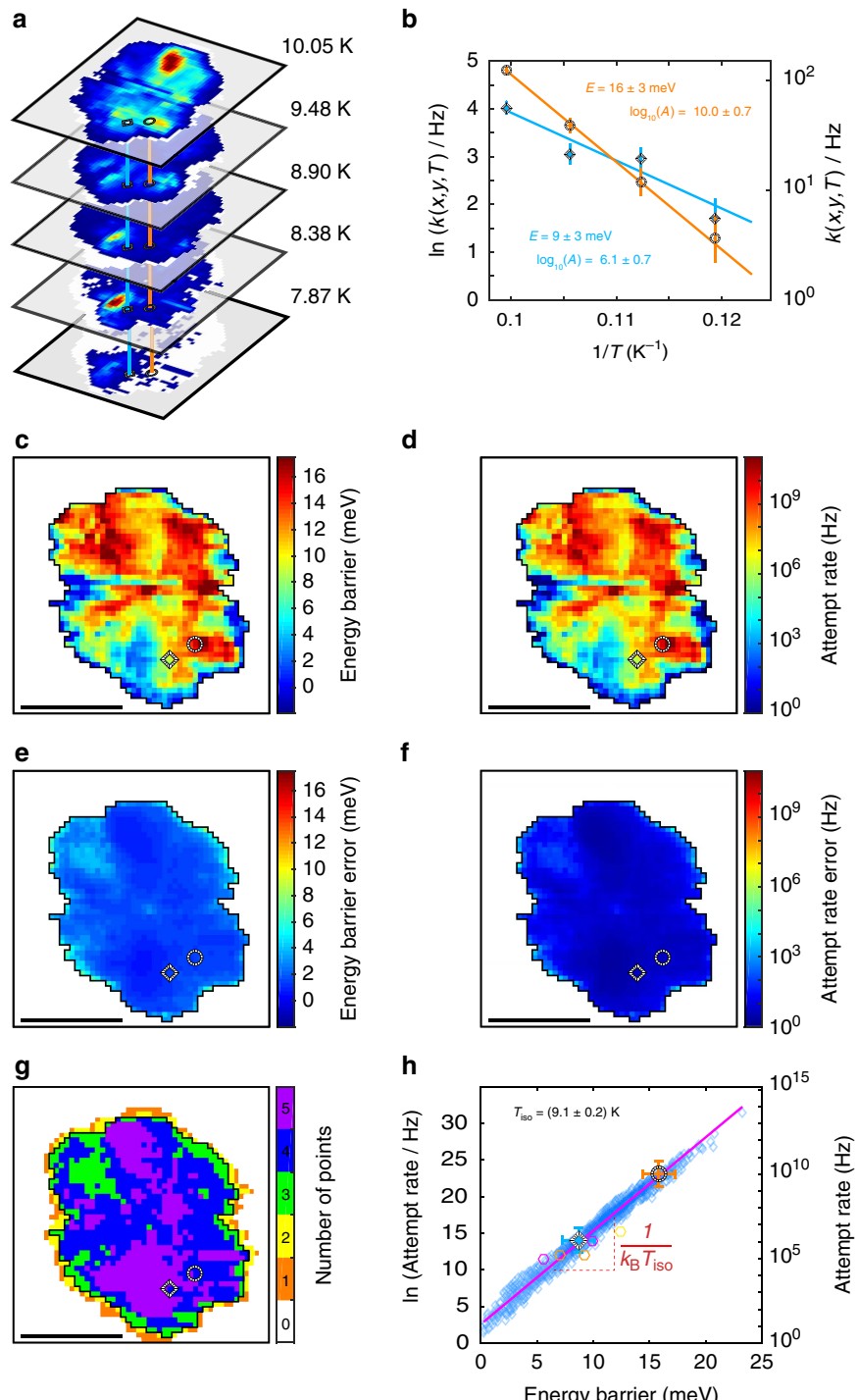

**Figure 3 | Arrhenius equation parameters from fits to the hopping rate k(x, y, T) of Figure 2b–g.** (**a**) Schematic of the procedure followed to obtain sets of position dependent hopping rates $k(x, y, T)$ for fitting with equation (1). The black/white striped ring and diamond markers identify two selected $(x, y)$ positions across temperatures. (**b**) Arrhenius plots of $k(x, y, T)$ for the two positions corresponding to the striped markers in (**a**) and Fig. 2. (**c,d**) Apparent energy barrier $E^*(x, y)$ and apparent attempt rate $A^*(x, y)$ from fitting equation (1) to each $(x, y)$ hopping rate temperature series. (**e,f**) Error of the measurement of (**c,d**), respectively. (**g**) Number of points in the temperature series contributing to position-dependent hopping rates for each $(x, y)$. (**h**) Plot of all pairs of $\ln(A^*(x, y))$ and $E^*(x, y)$ from (**c,d**) (blue diamonds), and the line fit (magenta line) showing the compensation effect between $E^*$ and $A^*$ from which an isokinetic temperature $T_{iso}$ follows. Values for fcc and hcp adsorption sites obtained from literature are indicated by hexagons in yellow[21] (fcc), magenta[23] (fcc) and orange[23] (hcp), respectively. All errors in the figure are s.d. Scale bars in (**c–g**), 1 nm.

*et al.*[27]), then a concurrent change in entropy would follow. However, it would not *a priori* follow that the entropy change should compensate the enthalpy (Fig. 3h and maps 4a,c).

That this compensation nevertheless takes place in many instances was discussed by Yelon *et al.*[28], who placed its origin in

multi-excitation entropy. Their model applies to cases where energy barrier is much larger than the typical thermal fluctuation energies available, a condition which is given for our experiment. In essence, a transition necessitates a potentially large number of small-energy excitations (of order $k_B T \approx 0.5$ meV) to co-occur, so

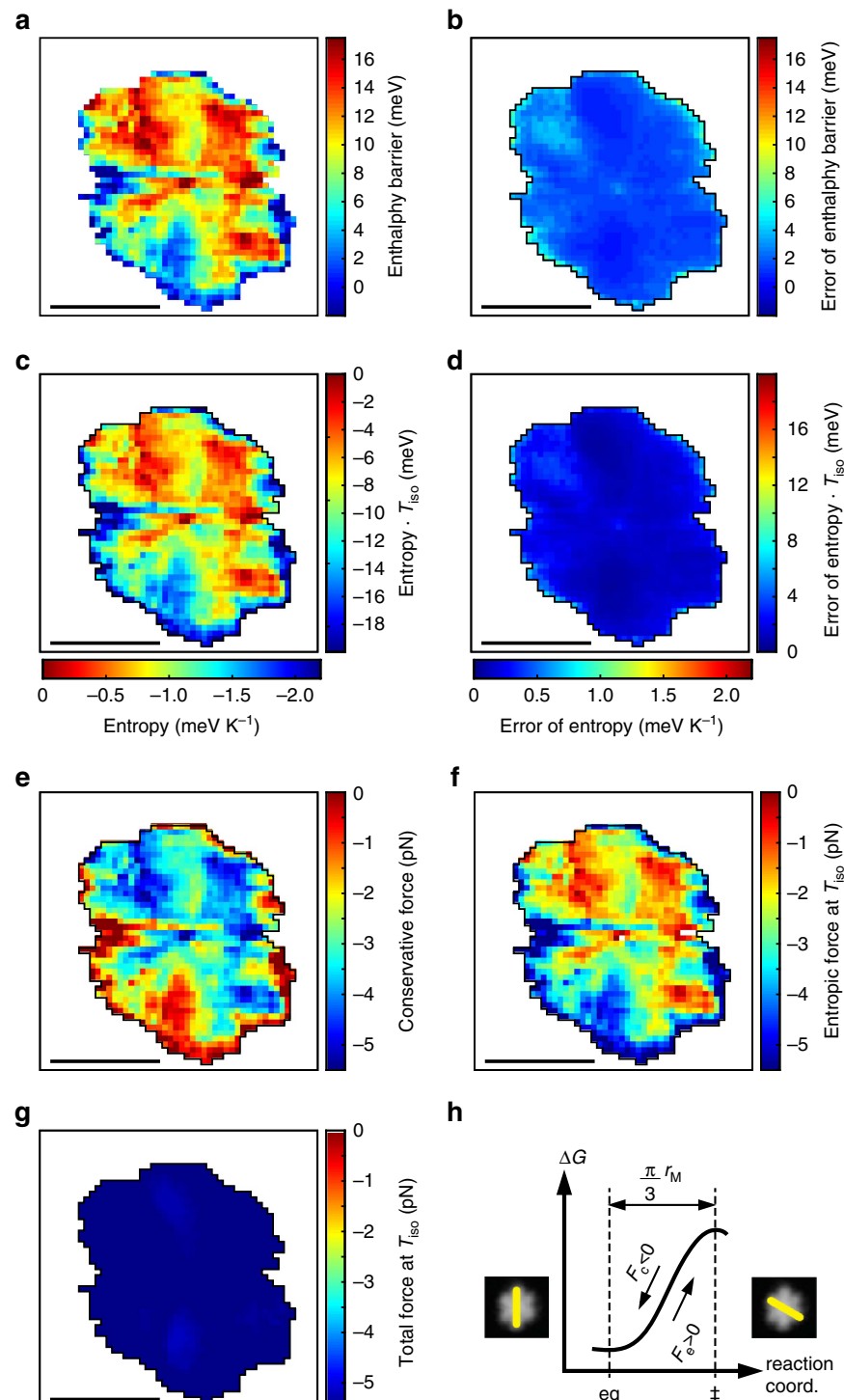

**Figure 4 | Enthalpy, entropy and derived conservative and entropic forces derived from the data in Figure 2b–g.** (**a**) Map of the enthalpy change $\Delta H^{\ddagger}$ obtained by fitting the rate maps from Fig. 3a with the logarithmic form of equation (1) using equation (5) for the (weakly) temperature dependent attempt rate, and $\Delta H^{\ddagger}(x, y) \equiv E^{*}(x, y)$. (**b**) Corresponding error. (**c**) Entropy difference $\Delta S^{\ddagger}$ from the fit of (**a**). The right hand scale bar is a scale of the corresponding thermodynamic potential at $T_{iso}$, which can be compared with (**a**). (**d**) Corresponding error. (**e**) Conservative force, derived from (**a**). (**f**) Entropic force at $T_{iso}$, derived from (**c**). (**g**) Sum of conservative and entropic forces at $T_{iso}$, from (**e**,**f**). (**h**) Schematic of the free energy change along the transition axis, indicating the sign convention for the conservative, $F_c$, and entropic, $F_e$, forces on the molecule. All errors in the figure are s.d. Scale bars in (**a**–**g**), 1 nm.

that the barrier is overcome. Yelon *et al.* then argue that a large number of excitations may be collected in a large number of ways and that this multiplicity gives rise to entropy[29].

From this perspective, any change in the barrier height induced by the tip would be followed by a compensation of the multi-

excitation entropy. It would be observed when other contributions to entropy do not dominate, and thus especially at low temperatures.

Last, comparing the conservative forces acting on the molecule with the emergent entropic forces shows the latter must be

included in correct descriptions of the transition dynamics, even at low temperatures. To see that, we can take the apparent barrier height $\Delta H^{\ddagger}$ and the potential $T_{iso}\Delta S^{\ddagger}$, and divide them by an effective distance $r_M \times \pi/3 \simeq 0.5$ nm, where $r_M = 0.5$ nm approximates half the length of the molecule. The resulting conservative and emergent (entropic) force maps, shown in Fig. 4e,f span a comparable range of $-5.5$–$0$ pN. These values are all about 100 times smaller than those required to push atoms across surfaces[30,31], which can be understood by the relative weakness of the van der Waals interaction between the butyl group and the Au(111) surface compared with metallic bonds. Because the values are much smaller, thermal fluctuations play a dominant role in the transitions.

At $T = T_{iso}$ the sum of the force maps in Fig. 4e,f is approximately independent of tip position (Fig. 4g), which illustrates the compensation of enthalpy and entropy.

## Methods

**Sample preparation.** We deposit dibutyl sulfide molecules at about 20 K on a Au(111) substrate mounted on a cooled manipulator, and subsequently transfer the sample to our home-built, low-temperature UHV-STM/SFM system located inside a bath cryostat. The temperature remains below 50 K, where molecular surface diffusion is suppressed sufficiently for many molecules to remain conveniently isolated (cf. Fig. 1a).

**Temperature stability during the STM measurement.** Temperature stability is a common experimental difficulty, which acquires particular relevance in our transition rate measurements, in which temperature must be approximately constant over several hours during which rate data is acquired. To ensure that the temperature does not change over the course of one measurement we rely on two temperature sensors. One is a diode sensor (Lakeshore, DT-470), which is accurate to within 12 mK at 10 K. The other is a Cernox sensor (Lakeshore model CX-1050-SD-HT 1.4L-P) with a warranted accuracy of 3 mK at 4.2 K and 6 mK at 10 K (9 mK at 20 K). The system regulates the temperature (Lakeshore 340 Temperature Controller) of the diode, placed on a cone-shaped copper block on the upper part of the microscope to within 0.1 mK (relatively). The lower part of the microscope is hanging in UHV inside thick Cu LHe-shields attached to the LHe reservoir. The sample-side temperature sensor is the aforementioned cernox sensor, which is located 130 mm away from the copper block, in the sample receiver, at about 13 mm distance from the sample. Its temperature is not regulated, but the measurements are carried out after at least 2.5 h of thermalization, when both cernox and diode sensors display the same temperature to within 3 mK. The temperature readings of either sensor do not change over the measurement. The first measurement in the temperature series is carried out at the lowest temperature after remaining at lowest temperatures for at least two days, and the sample is heated in subsequent measurements. We take the temperature error propagated in the fits providing attempt-rate and enthalpy barriers to be at most 3 mK.

**Scan conditions.** For all STM imaging in this work we use a bias of 200 mV and keep the (average) tunnelling current constant at 20 pA, which amounts to a tunnelling resistance of 10 G$\Omega$. It is generally believed that the tip does not influence the dynamics of the molecule under such tunnelling conditions[21,26].

**Transition rate data analysis.** In contrast to prior work[21,23], high resolution data ($\approx 50$ pm square per pixel) are taken over a 5.2 nm × 3.3 nm area at six different temperatures ($T_{i=1,...,6} = 7.39, 7.87, 8.38, 8.9, 9.48$ and $10.05$ K) using a slow z-feedback (feedback parameters: $P = 10^{-5}$ nA nA$^{-1}$, $I = 1$ ms) at a scan rate of 400 s per line (2 s per pixel in each of the forward/backward direction). This leads to 100 × 64 point topographs (for example, Fig. 1c) coupled with 3,000 point tunnel current traces (for example, Fig. 2a) sampled at 1.5 kHz, beyond the bandwidth of our current preamplifier (1.1 kHz for a current-to-voltage conversion gain of $10^9$ V A$^{-1}$). A posteriori analysis of this latter data allows us to obtain at each pixel the number of hops $N(x, y, T)$, identified as the discrete jumps in the tunnelling current error. The hopping rate is simply this divided by the time spent on each pixel, that is, $k(x, y, T) \equiv N(x, y, T)/2$s.

The scan range of the piezo scanner increases with temperature and can lead to small changes in the scale of the images. We account for this fact with separate calibration measurements over the relevant temperature range (not shown) based on fiduciary marks on $\approx 40$ nm × 40 nm images of the Au(111) surface topography.

It is also necessary to align the images to correct the drift between data sets acquired at different temperatures. We accomplish this by means of a plain phase correlation[32] algorithm implemented in Matlab using current error signal ($I(t) - 20$ pA, not shown), but nearly identical results are found using the

topography images. The reference image for registration is the one we measured at 8.9 K, close to the middle of the series of temperature images.

**Compensation of apparent energy barrier and -attempt rate.** The compensation of transition state entropy and enthalpy results from fits to a set of about 1,000 temperature dependent rate data-sets. The statistical error of the count (of order square root of the count) dominates other sources of error and ultimately results in the bounds indicated for each tip-position's enthalpy and entropy. We can exclude common interpretation mistakes[33,34] by confirming that the rate patterns show tip-position dependent lobes which evolve continuously with temperature. This spatial correlation implies that the measurements are not random. We also necessarily measure rates during slow scans. The rates obtained for two successive points in one scan line are about 2 s apart. We thus have two distinct rate measurements at points being only 50 pm apart. Both yield essentially the same rate pattern with tip position and temperature dependence, which is inconsistent with random errors. The same can be said from data from consecutive lines, which describe approximately the same points, that is, to a precision of a pixel width of 50 pm. Two successive scan lines are about 400 s apart. In addition, there are significant differences among many (most) $k(1/T)$ rate-lines, which cannot be ascribed to random error. Finally, the analysis of a second molecule with the same method produces similar results, as we show in Supplementary Fig. 1. Specifically, as discussed in Supplementary Notes 1, we reobtain the compensation line, although the lobe structure is different, but shares with the former that it lacks the symmetry of the high-temperature topography of the molecule, and departs from it in a similar way.

**Data availability.** The rate-data supporting the findings of this study is available at http://osf.io/p7arh

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

## Acknowledgements

Support from the Swiss National Science Foundation (Sinergia CNRSII2 136287/1), and Empa is hereby gratefully acknowledged. We thank S.M. Vranjkovic for strong support in designing and constructing the STM utilized in this study, and acknowledge financial support from the EU FP7 program NMP-2007-1.2-2 (grant 214250). We thank K.-H. Ernst for supplying the molecules and various scientific discussions on molecular motion on surfaces, as well as A. Baratoff for detailed and helpful comments.

## Author contributions

J.C.G. and M.Pe. carried out the measurements, and set up the STM together with M.Pa., E.W.H., J.S. and H.J.H. J.C.G., M.Pe., M.A.M., H.J.H. and E.W.H. analysed the results. M.A.M., H.J.H. and E.W.H. wrote the manuscript. All authors discussed and commented the manuscript.

## Additional information

**Competing financial interests:** The authors declare no competing financial interests.

**Publisher's note**: 

