## [Peer Review File · Nature Communications]

Reviewers' Comments:

Reviewer #1 (Remarks to the Author)

The authors show that when fitting rate data for single molecule transitions of moieties adsorbed on a surface as measured by STM to an Arrhenius law, the frequency factor normally taken to be a constant, depends very significantly on the position of the STM tip relative to the adsorbed molecule. This can dramatically confound interpretation of the temperature dependence of the transitions. Because the difference between the activation enthalpy divided by temperature and the activation entropy can be very small, the observed behavior for the transition dynamics is very sensitive to small variations in experimental tip position, etc. even if the activation enthalpies themselves are quite large.

Overall this is a very useful paper that points to possible origins of inconsistencies in the literature for temperature dependence of hopping dynamics of surface adsorbed molecules. I think that the paper will stimulate further research, particularly in systems for which discrepancies between reported data exist, that will lead to significant refinements of insights about the behavior surface adsorbed molecules and their transitions. Consequently I recommend publication.

R. Dean Astumian

Reviewer #2 (Remarks to the Author)

Report on "Surface single-molecule dynamics controlled by entropy at low temperatures"

This manuscript reports on detailed temperature dependent measurement and thorough analysis of the motion of DBS molecules on Au(111). The analysis goes far beyond anything I have seen in the literature, and shows how quantitative analysis on these systems can be accomplished to yield important parameters related to activation energies, attempt frequencies, and entropy of molecule-surface system. I recommend publication as is in Nature Comm.

Some minor typos, line 168, I think Fig. 3F should be 3H?

Fig. 4D, this error image is not near zero according to the scale bars.

Reviewer #3 (Remarks to the Author)

In the work of Gehring et al. with the title "Surface single molecule dynamics controlled by entropy at low temperatures" the authors report on a detailed scanning tunneling microscopy (STM) study of the dynamics of individual dibutyl-sulfide (DBS) molecules on a Au(111) surface depending on the sample temperature as well as of the STM tip position. The acquired data is quite detailed and impressive. The DBS molecule is anchored via the sulfur to the Au(111) surface but able to rotate around the corresponding bond thermally induced at sufficient temperatures. The authors are able to perform isothermal measurements at low temperatures ($\sim 5-10$ K) and to estimate the frequencies of the DBS movement. Based on this data a corresponding Arrhenius analysis was conducted yielding values for A (referred to as attempt frequency by the authors) and E (activation energy). After finding a compensation effect of the latter parameters the Eyring formalism is used to extract the corresponding entropy values S and enthalpy values H. Again the authors find that there is a compensation effect.

Even though the data base is very nice I see major flaws in the discussion and interpretation of the data. First of all I cannot comprehend why the authors refer to the expression: attempt frequency for A (instead of preexponential factor also commonly used)? This expression is derived from collision theory and is not suitable for the discussed problem, since this number is indeed not tied to a physical process with a certain frequency as can be easily seen by equation 4 in the paper (differences in entropy from initial to transition state also account). Along that line I am also not sure why the authors use Arrhenius analysis in the first place and after that a similar analysis

following Eyring's formalism. It is very clear that Arrhenius is an empirical formula without direct theoretical background (in addition E is not a well-defined energy in thermodynamics), in that sense the Eyring formalism is certainly the right choice since here a solid theoretical background along with real thermodynamic potentials is provided. However, from my point of view the authors again missed to introduce the Eyring formalism in the most comprehensible way (c.f. equation 3 in manuscript) which would be with the free Gibbs energy G in the exponent (this was also Eyring's starting point). G finally determines the activation barrier for a process within transition state theory and is consequently the determining value. In this framework G is given by the Helmholtz equation: $\Delta G = \Delta H - T \Delta S$. In this context the similarities in Fig 4 (A) and (C) just mean that the activation barrier given by the corresponding free Gibbs energy results in a constant but very low energy value (around 1 meV?). Therefore one of the major findings given in figure 4 (G) just mean that ΔG is constant probably on a very low level (why do the authors not stick to energy values but go for forces??). From my perspective the at least almost constant ΔG is a very important point and should be discussed in detail.

Another crucial point, which is completely missing, is how the observed patterns for the different thermodynamic potentials relate to the symmetry (atomic order) of the substrate?...in particular to the underlying herringbone structure. In this respect it is also clear that the structure of the underlying substrate basically determine the potential energy landscape. To me it appears quite natural that with changing potential also the partition function and thus the entropic contribution changes. Maybe the authors can find a corresponding correlation between the depicted patterns and the atomic order of the substrate. In this regard I would strongly recommend checking figure 1 of the supplementary information. Here it appears that exactly the symmetry of the substrate is reproduced by the data. I rather think that this data set is the one to show and discussed maybe with a focus on the potential energy landscape of the underlying substrate. In this picture the STM would indeed have no influence on the transition process, which is also something I would expect given the tunneling resistance of 10 GΩ. I believe if the authors would get theoretical support, this perspective might become even more meaningful. The result would be still extraordinary, since this is probably the first case where one can spatially follow up on the transition coordinate on the molecular level. In this light also the title should be changed.

Other points which I would like to see addressed:

-What is the mean variation of the frequency values depicted in figure 3 (D) and the corresponding average value of the corresponding errors given in figure 3 (F)? This is very difficult to extract, since the color scale in figure 3 (F) makes it hard to extract the values. I anticipate that the error is quite large in comparison of the mean variation of the frequency values. This is also confirmed by the plot of the blue line fitted to the data points in figure 3 (B). Three of the four values are hardly on the fitted line even considering the error bars. As a consequence the slope of the fitted line is given with an error of one third of the estimated value (9 plusminus 3). My understanding is, that the also given values for $\log(A)$ (6.1 plusminus 0.7) are extracted by determining the ordinate intercept. I tried to do that graphically in a rough manner and yielded a more than doubled value (~ 13) and also an much larger error...maybe the authors can comment on this issue.

-What is the reason that the authors give measurements at five different temperatures in figure 3 (A) but only have four points to fit the lines in Fig 3 (B)? This is even more severe in figure 1 of the sup info where they show eight data sets for different temperatures but again only 4 points to fit the lines. I am aware of the fact that for the lower temperatures some points the frequency is zero, but still one could use the data points with frequencies larger than zero and check if there is still Arrhenius behavior for that tip positions.

-Concerning the comment on pg 3 starting at the top of the page: there are also numerous examples where the frequencies are too high, e.g.: Roos et al., Phys.Chem.Chem.Phys, 2010,12,818 and Campbell et al., J.Am.Chem.Soc, 2012,134,18109.

In the light of my comments above I do not find the manuscript suitable for NATURE COMMUNICATIONS in its present form. From my perspective the discussion and interpretation must be reconsidered in a major way. I am convinced that the interpretation as is, is not fully

comprehensible (what exactly is the influence of the tip and how does it change the transition state?). Instead, I believe that the entropy and enthalpy indeed spatially changes (also without the spectator STM tip!) due to the underlying surface structure practically realizing the corresponding potential energy landscape for the transition process. In particular with theoretical support the story might become a different twist (correlation with substrate) and might be publishable in NAT COMMUN after major revision.

Authors' reply to comments on "Surface single-molecule dynamics controlled by entropy at low temperatures"

Reviewers' comments:

Reviewer #1 (Remarks to the Author):

The authors show that when fitting rate data for single molecule transitions of moieties adsorbed on a surface as measured by STM to an Arrhenius law, the frequency factor normally taken to be a constant, depends very significantly on the position of the STM tip relative to the adsorbed molecule. This can dramatically confound interpretation of the temperature dependence of the transitions. Because the difference between the activation enthalpy divided by temperature and the activation entropy can be very small, the observed behavior for the transition dynamics is very sensitive to small variations in experimental tip position, etc. even if the activation enthalpies themselves are quite large.

Overall this is a very useful paper that points to possible origins of inconsistencies in the literature for temperature dependence of hopping dynamics of surface adsorbed molecules. I think that the paper will stimulate further research, particularly in systems for which discrepancies between reported data exist, that will lead to significant refinements of insights about the behavior surface adsorbed molecules and their transitions. Consequently I recommend publication.

Authors' comment

We thank the reviewer for the comments and highly value the recognition of our experimental results' significance. We very much look forward to the research triggered by this work. Thank you.

Reviewer #2 (Remarks to the Author):

Report on "Surface single-molecule dynamics controlled by entropy at low temperatures"

This manuscript reports on detailed temperature dependent measurement and thorough analysis of the motion of DBS molecules on Au(111). The analysis goes far beyond anything I have seen in the literature, and shows how quantitative analysis on these systems can be accomplished to yield important parameters related to activation energies, attempt frequencies, and entropy of molecule-surface system. I recommend publication as is in Nature Comm.

Some minor typos, line 168, I think Fig. 3F should be 3H?

Fig. 4D, this error image is not near zero according to the scale bars.

Authors' comment

We thank the reviewer for the comments and for recognizing the quality of the measurements and analysis. Thank you also for pointing out the typo in line 168 and especially for spotting the scale bar mistake in Figure 4D, which we had completely overlooked. Thank you.

Reviewer #3 (Remarks to the Author):

Reviewer comment:

In the work of Gehring et al. with the title “Surface single molecule dynamics controlled by entropy at low temperatures” the authors report on a detailed scanning tunneling microscopy (STM) study of the dynamics of individual dibutyl-sulfide (DBS) molecules on a Au(111) surface depending on the sample temperature as well as of the STM tip position. The acquired data is quite detailed and impressive. The DBS molecule is anchored via the sulfur to the Au(111) surface but able to rotate around the corresponding bond thermally induced at sufficient temperatures. The authors are able to perform isothermal measurements at low temperatures (~5-10 K) and to estimate the frequencies of the DBS movement. Based on this data a corresponding Arrhenius analysis was conducted yielding values for A (referred to as attempt frequency by the authors) and E (activation energy). After finding a compensation effect of the latter parameters the Eyring formalism is used to extract the corresponding entropy values S and enthalpy values H. Again the authors find that there is a compensation effect.

Authors' comment

We thank the reviewer for these and the other detailed comments below, as well as for the appraisal of the acquired data. As far as we know this is the first time an enthalpy-entropy compensation effect has been found in a system comprising only one transitioning molecule – normally, compensation effects require a comparison of the temperature-dependent transition rates for systems with e.g. different chemistry (for instance the elemental composition of a metal cluster in V.N. Antonov *et al.*, *Phys. Rev. B* **70** (2004) 045406, or the type of alkane in catalysis reactions in G. C. Bond *et al.*, *Catalysis Reviews* **42** (2000) 323).).

Reviewer comment:

Even though the data base is very nice I see major flaws in the discussion and interpretation of the data. First of all I cannot comprehend why the authors refer to the expression: attempt frequency for A (instead of preexponential factor also commonly used)? This expression is derived from collision theory and is not suitable for the discussed problem, since this number is indeed not tied to a physical process with a certain frequency as can be easily seen by equation 4 in the paper (differences in entropy from initial to transition state also account). Along that line I am also not sure why the authors use Arrhenius analysis in the first place and after that a similar analysis following Eyring's formalism. It is very clear that Arrhenius is an empirical formula without direct theoretical background (in addition E is not a well-defined energy in thermodynamics), in that sense the Eyring formalism is certainly the right choice since here a solid theoretical background along with real thermodynamic potentials is provided.

Authors' reply:

Thank you for the appraisal of the data. We disagree that there are any major flaws in the analysis, but take this as a good opportunity to readjust the way we explain our findings.

The reviewer correctly pointed out that ‘pre-exponential factor’ is “*also commonly used*” besides the term ‘attempt rate.’ Implicit is the notion that none of the terms is used exclusively, and so this is a matter of preference. We had chosen ‘attempt rate’ for its meaning of probability amplitude, which is not specific to collision theory, nor is it used exclusively in that theory (in fact, we all agree that the Arrhenius law is phenomenological.) This meaning of probability amplitude (probability of rotational transition in this case) is better conveyed by ‘attempt rate’ than by ‘pre-exponential factor’, we think.

We have decided to explicitly use the pre-exponential factor terminology preferred by the reviewer in the abstract and introduction, lest any possible connection with collision theory is too strongly perceived, but we retained several instances of ‘attempt rate’ in turn.

The “Arrhenius analysis” is often used in literature of research in the field (which we cite), so we do not think this will throw the reader off track. Of course, it clearly is a phenomenological expression which allows us to identify parameters that do not depend (in first instance) of the temperature variable, namely the effective barrier and pre-factor or attempt rate. Because Arrhenius ‘laws’ are phenomenological, using one has the advantage that it does not require assuming a priori any particular temperature dependence for the pre-exponential factor. It thus seems a rather general approach when one is developing an understanding of the system’s physics.

The phenomenological Arrhenius approach establishes the characteristics of the data, which are fact, independently of any theory and before these issues need to be analyzed in order to develop the understanding of the system.

On the basis of our results we ultimately agree with the reviewer that using the Eyring formalism is convenient. But our results also indicate that although there is a theoretical background to this formalism, it is not at all evident how to calculate the involved thermodynamic potentials. Specifically, it is debatable how conformational and excitation entropy contribute to the entropy that must be used in Eyring’s equation, or how substrate and tip affect the potential landscape and the entropy (for example, this paper shows that the tip has an influence, which was unexpected).

Reviewer comment

However, from my point of view the authors again missed to introduce the Eyring formalism in the most comprehensible way (c.f. equation 3 in manuscript) which would be with the free Gibbs energy G in the exponent (this was also Eyrings starting point). G finally determines the activation barrier for a process within transition state theory and is consequently the determining value. In this framework G is given by the Helmholtz equation: $\Delta G = \Delta H - T \Delta S$.

Authors’ reply:

We think the expression in Eq. (3) of the original manuscript does make a clear connection with the variability of A , inasmuch as it is a sum over possible microstates, and this number thereof is directly linked to S . Nonetheless, we have reworked the text, essentially swapping Eqs. (3) and (4) of the original manuscript, and rewording the text accordingly. We hope this presentation of the equation is clearer.

Reviewer comment

In this context the similarities in Fig 4 (A) and (C) just mean that the activation barrier given by the corresponding free Gibbs energy results in a constant but very low energy value (around 1 meV?). Therefore one of the major findings given in figure 4 (G) just mean that delta G is constant probably on an very low level (why do the authors not stick to energy values but go for forces???). From my perspective the at least almost constant delta G is a very important point and should be discussed in detail.

Authors' reply:

To clarify: the low, uniform value of ΔG is found (*only*) at one specific temperature, as we discuss below. We think that a detailed discussion of ΔG , which in fact constitutes the whole latter part of the manuscript, is best held by disaggregating the enthalpic and entropic contributions to the free energy, as we do, because this explicitly introduces the main temperature dependence of ΔG , which we investigate experimentally.

In fact, because ΔG is temperature dependent, and because that temperature dependence is primarily given by the Helmholtz equation $\Delta H - T \Delta S$ (as the reviewer pointed out), in which ΔH and ΔS are approximately temperature independent (by virtue of being well-fitting Arrhenius-type parameters to the temperature dependent transition rate data), one will not find a uniform ΔG map for all temperatures unless ΔH and ΔS are themselves uniform. They are clearly not, but neither are they unrelated. The relation between both can be described by two equivalent conditions: that 1), ΔH and ΔS are linearly related (compensation law); or that 2), there exists a single temperature for which all rates are equal (the isokinetic temperature). We measure ΔH and ΔS for a map of relative tip-sample positions, so condition 1) implies that maps for ΔH and ΔS have to be the same up to a scale factor. This is what we find. Conversely, condition 2) means that all the different pairs of ΔH and ΔS (obtained for the different tip-molecule positions covered in the scan) give rise to the same ΔG at one particular temperature, at which therefore the transition rate becomes the same for all relative tip-sample positions. This is the meaning of Figure 4G, if we dispense of the proportionality factor relating the thermodynamic potential with a thermodynamic driving force.

But concerning the use of forces, it is useful to allow a direct comparison with forces encountered during atomic manipulation, and confirm that we measure forces that are much smaller than those. Moreover, such forces are, at least in principle - measurable with a suitable scanning force / tunneling microscopy combination.

Reviewer comment

Another crucial point, which is completely missing, is how the observed patterns for the different thermodynamic potentials relate to the symmetry (atomic order) of the substrate?...in particular to the underlying herringbone structure. In this respect it is also clear that the structure of the underlying substrate basically determine the potential energy landscape. To me it appears quite natural that with changing potential also the partition function and thus the entropic contribution changes. Maybe the authors can find a corresponding correlation between the depicted patterns and the atomic order of the

substrate. In this regard I would strongly recommend checking figure 1 of the supplementary information. Here it appears that exactly the symmetry of the substrate is reproduced by the data. I rather think that this data set is the one to show and discussed maybe with a focus on the potential energy landscape of the underlying substrate. In this picture the STM would indeed have no influence on the transition process, which is also something I would expect given the tunneling resistance of 10GOhm.

Authors' reply:

Yes, the substrate evidently determines the features of the potential energy landscape for the molecule if the tip is not present. As discussed, our measurement conditions (10GOhm tunnelling resistance, 200mV bias) are considered non-perturbing, and one might expect the rotation dynamics to be unaffected by the tip.

At sufficiently high temperatures (see Fig. 1G in our manuscript) the molecule hops between all the adsorption sites. All the hops are counted by the tip when it is placed above the molecule because we do not discriminate between different amplitudes of the current level change upon hopping. So if the rotation dynamics were indeed unaffected by the tip, we would measure a homogenous rate pattern for tip positions above the molecule. Now, while our rate measurements do show a finite hopping rate (particularly at higher temperatures for all tip positions, where non-zero counts shape an area of roughly hexagonal shape), they strongly deviate from a 'single' homogeneous level for different tip positions. This shows that the rate depends on the position of the tip.

Hence, although the surface surely defines the adsorption sites of the molecule and main rotation dynamics, the tip does influence the latter. Therefore, the origin of the lack of symmetry of the patterns can be traced back to the structure of the tip with which the molecule interacts.

We thank the referee for commenting on these issues, which may not have been as clear in our original manuscript as we would have hoped. The revised manuscript (see explanation beginning at line 120) addresses the above points explicitly.

Concerning the existence of a relation between the partition function and the potential, the reviewer is correct. Different potential energy landscapes between equilibrium and transition state may define correspondingly different entropy changes between these states. However, we would not necessarily expect a linear relationship between enthalpy and entropy changes to hold over orders of magnitude, which is the essence of the compensation law that we find for this molecule. This is not trivial. When the linear dependence is given then enthalpy and entropy are compensated. We show this is the case in our example. It also occurs in other groups of systems, but to our knowledge has never before been observed in systems comprising a single molecule. So far (to see if there is compensation) it was necessary to e.g. modify the type of material system (for instance the elemental composition of a metal cluster in V.N. Antonov *et al.*, *Phys. Rev. B* **70** (2004) 045406, or the type of alkane in catalysis reactions in G. C. Bond *et al.*, *Catalysis Reviews* **42** (2000) 323).

The fundamental question we see is: how does compensation come to be in this system? We mention the multi-excitation entropy as an intriguing concept that could *possibly* provide insight into that question. As we show in this work, the STM tip can be used to analyze and select the transition conditions in a single molecule, which is itself very interesting.

Reviewer comment

I believe if the authors would get theoretical support, this perspective might become even more meaningful. The result would be still extraordinary, since this is probably the first case where one can spatially follow up on the transition coordinate on the molecular level. In this light also the title should be changed.

Authors' reply

We would like to qualify our overall agreement with the reviewer's statements:

We will agree with the reviewer that our work opens the way to following a transition on the molecular level. We are also convinced that theoretical work will provide insight into the meaning of entropy in these systems and its influence on the transitions, and enthusiastically support these efforts. Such work must begin to consider the system to be comprised of substrate, molecule and tip, and provide physical descriptions e.g. for thermal fluctuation energy transfers within the system, especially to the extent that they are fundamental for multi-excitation entropy. In some simple cases fundamental models might be computationally accessible. But all that work would far exceed the scope of this article, and even assuming adequate theoretical support could be developed in a short time, we strongly disagree that it should be a prerequisite for publication.

Concerning the title of the manuscript, we think that whereas the tip may have been considered to affect the potential seen by the molecule, the possibility that it thereby also modifies the entropy on a sub nm scale was unexpected. So although the comment of the reviewer has merit, in our current opinion the title is adequate for this work and could be retained without detrimental effects on the paper.

Reviewer comment

Other points which I would like to see addressed:

What is the mean variation of the frequency values depicted in figure 3 (D) and the corresponding average value of the corresponding errors given in figure 3 (F)? This is very difficult to extract, since the color scale in figure 3 (F) makes it hard to extract the values. I anticipate that the error is quite large in comparison of the mean variation of the frequency values. This is also confirmed by the plot of the blue line fitted to the data points in figure 3 (B). Three of the four values are hardly on the fitted line even considering the error bars. As a consequence the slope of the fitted line is given with an error of one third of the estimated value (9 plusminus 3). My understanding is, that the also given values for $\log(A)$ (6.1 plusminus 0.7) are extracted by determining the ordinate intercept. I tried to do that graphically in a rough manner and yielded a more than doubled value (~ 13) and also an much larger error...maybe the authors can comment on this issue.

Authors' reply

The mean value of $\log_{10}(A/\text{Hz}) = 6.43$ and the mean of its error is 1.06. We agree with the referee that these values are difficult to extract from images, but note that the maps of the energy barrier, $\log_{10}(A/\text{Hz})$, and those of the corresponding errors are plotted with the *same color scale*. The fact that energy barrier and $\log_{10}(A/\text{Hz})$ lobe patterns appear with colors ranging from (high value) reds to (low value) blues while the corresponding error maps show weak variation in the dark blue color range shows that the determined energy barriers and $\log_{10}(A/\text{Hz})$ -values, and the observed variations of their patterns are statistically significant.

The reviewer is correct that A is the fitted intercept of the plot of $\ln(\text{rate})$ v. $(1/T)$ that fits the data (at this stage; Delta S is a different fit that includes the weak T dependence from $k_B T/h$). I repeated the fit graphically to reproduce your finding (specifically, I overlaid rectangles e.g. in Powerpoint and compared their dimensions with the plot scale), and I obtain 6.2 and 10.1, which is about equal to the fitted results in the two representative examples. But (other than the base in the logarithm) the only reason for the discrepancy between your estimate and mine that comes to mind is the logarithmic amplification for the intercept.

The errors are discussed in detail in section III of the supporting material, but to summarize, the main source of error is the statistical error of the count, which is ultimately the square root of the count. The examples were chosen to display very different values of E^* and A^* , and correspond to points not too far apart but located in clearly different parts of the lobe structure. We believe the quality of the fit of this and the about 1200 other points is more easily gauged from Figures 3E and F, which show that the precision of our measurement is sufficient to reveal the effect we want to highlight.

At any rate, going over the figures we noticed the unintended labeling of the error of the map in Figure 3D, depicted in Figure 3F, as “A*” instead of “attempt rate error”. We have corrected this in the revised manuscript.

Reviewer comment

What is the reason that the authors give measurements at five different temperatures in figure 3 (A) but only have four points to fit the lines in Fig 3 (B)? This is even more severe in figure 1 of the sup info where they show eight data sets for different temperatures but again only 4 points to fit the lines. I am aware of the fact that for the lower temperatures some points the frequency is zero, but still one could use the data points with frequencies larger than zero and check if there is still Arrhenius behavior for that tip positions.

Authors' reply:

Thank you for the very precise observation. The rates obtained at the lower temperatures display a subtle but obvious lobe structure, and we show this to further support the results of the higher temperatures qualitatively. Tip positions with zero counts at the lowest temperatures indicate that the scanning time was insufficient to obtain a meaningful transition count (unfortunately it is not practical to use even slower scan times to obtain higher count numbers). As pointed out in Figure 2, we accordingly restricted the analysis to positions in which the temperature set of measurements included at least 3 points, reported in Figure 3G. The points depicted in Figure 3B are simply examples, as we describe above, which happen to comprise just 4 points.

Concerning Figure 1S, without exception all counts of the first temperature (5.42K) are minimal, so we do not think they represent the transition behavior quantitatively. We have consequently excluded this data set. Actually, a 5.45K even lower count measurement exists for the first molecule in figure 2 in the main text, but do not show it for space reasons. For the fits reported in Figure S1 K-O we again use at least 3 points, corresponding to the positions indicated in Figure S1 I. There is a small typographical error in that panel: the purple regions labeled as comprising exactly 5 points should actually be labelled as comprising 5 points or more (up to 8). Sorry – we corrected it. As for using 4 points in panel S1 J, this is again just a convenient example.

Reviewer comment

Concerning the comment on pg 3 starting at the top of the page: there are also numerous examples where the frequencies are too high, e.g.: Roos et al., Phys.Chem.Chem.Phys, 2010,12,818 and Campbell et al., J.Am.Chem.Soc, 2012,134,18109.

Authors' reply:

Thank you for bringing these references to our attention. The comment you refer to was meant to point out a disconnection between a phenomenological Arrhenius law and a mechanistic view of the transition process. This is known and may not require the emphasis we put on it, as the work of Roos *et al.* shows in the reference you share. We think it is best in the revised manuscript to reword the abstract without changing the meaning too much, from:

“This approach yields consistent energy barrier values, but also attempt rates orders of magnitude below expected oscillation frequencies of particles in the meta-stable state. Moreover, even for identical systems, the measurements can yield values differing from each other by orders of magnitude.”

To:

“This approach provides characteristic energy barrier and exponential pre-factor values, but even for identical systems, the measurement of these parameters can yield significantly different values. Pre-factors, in particular, can differ by orders of magnitude.”

Reviewer comment

In the light of my comments above I do not find the manuscript suitable for NATURE COMMUNICATIONS in its present form. From my perspective the discussion and interpretation must be reconsidered in a major way. I am convinced that the interpretation as is, is not fully comprehensible (what exactly is the influence of the tip and how does it change the transition state?). Instead, I believe that the entropy and enthalpy indeed spatially changes (also without the spectator STM tip!) due to the underlying surface structure practically realizing the corresponding potential energy landscape for the transition

process. In particular with theoretical support the story might become a different twist (correlation with substrate) and might be publishable in NAT COMMUN after major revision.

Authors' reply:

We disagree. Before addressing these final remarks in greater detail we want to stress that the connection between the potential landscape and the surface is not disputed and is a minor point in the paper, because ultimately we want to emphasize that entropy and enthalpy are compensated in this single molecule, and that we can use the tip to obtain a view of the molecule dynamics with sub-molecular precision.

Conventional wisdom would state that the tip has no effect on measurements in our case. *We disprove this*, as we elaborate above, but to improve our manuscript we have carefully reworked the sections that might have been unclear in this respect (For the benefit of the reviewers we highlighted all changes of the original text in blue).

Our work is an important first example where the “*exact... influence of the tip and how does it change the transition state*” can be *quantified* experimentally, at least for our measurement conditions (take e.g. the barrier map of Figure 3C). This is the first time that compensation is seen in a single molecule. Prior to this work, to modify the energy barrier (to see if there is compensation) it was necessary to e.g. modify the type of material system (for instance the elemental composition of a metal cluster in V.N. Antonov et al., Phys. Rev. B 70 (2004) 045406, or the type of alkane in catalysis reactions in G. C. Bond et al., Catalysis Reviews 42 (2000) 323).

Fundamental and highly interesting questions arise, concerning the mechanism by which the tip-influence is exerted and gives rise to the compensation law found. The example we have analyzed suggests different ways to approach the problem in future work. Specifically, by properly selecting the relative tip-molecule position we can select the point on the compensation line that we want to investigate. Possibly the tip's characteristics (and adsorption site on the herringbone, as also hinted by the reviewer) can be used to influence the slope of the compensation line.

Concluding remarks:

We would like to thank the reviewer for valuing our experimental data, for the criticism, the useful references, and spotting deficiencies in the manuscript that we could address thanks to it. As we explain, the perceived deficiencies brought up by the reviewer do not constitute real impediments to publication. We have taken the reviewer's comments to heart, nevertheless, and addressed in the revised text the discussion triggered by the reviewer.

We think that there is now no substantial obstacle to publication of these unique experimental results, and remain convinced of the importance that they have for future research, as accorded by reviewer 1 and 2. Thank you.

Reviewers' Comments:

Reviewer #3 (Remarks to the Author)

Gehring et al. responded to my review of their paper with the title "Surface single molecule dynamics controlled by entropy at low temperatures" with some modifications to their original manuscript and an elaborate rebuttal letter. Some of the points raised in my report are adequately addressed others from my point of view not. For example I still have problems to comprehend their arguments concerning the role of the tip and I also still think that the role of the substrate is trivialized. Even though not all conclusions might be fully correct in the revised form, the interesting paper might appeal to the broad readership of NATURE Communications and can be released as is.